# Exploring the Antibacterial Potential of Lamiaceae Plant Extracts: Inhibition of Bacterial Growth, Adhesion, Invasion, and Biofilm Formation and Degradation in *Pseudomonas aeruginosa* PAO1

**DOI:** 10.3390/plants13121616

**Published:** 2024-06-11

**Authors:** Mariana Oalđe Pavlović, Stoimir Kolarević, Jelena Đorđević Aleksić, Branka Vuković-Gačić

**Affiliations:** 1University of Belgrade—Faculty of Biology, Institute of Botany and Botanical Garden “Jevremovac”, Studentski Trg 16, 11000 Belgrade, Serbia; brankavg@bio.bg.ac.rs; 2University of Belgrade—Institute for Biological Research “Siniša Stanković”, National Institute of the Republic of Serbia, Bulevar Despota Stefana 142, 11000 Belgrade, Serbia; stoimir.kolarevic@ibiss.bg.ac.rs; 3University of Belgrade—Institute for Multidisciplinary Research, Kneza Višeslava 1, 11000 Belgrade, Serbia; jelenadjo@imsi.rs

**Keywords:** Lamiaceae extracts, MIC, *Pseudomonas aeruginosa* PAO1, bacterial invasion, biofilm formation, biofilm degradation

## Abstract

In response to the global rise in antibiotic resistance and the prevalence of bacterial biofilm-related infections, the antibacterial efficacy of methanolic, ethanolic, and aqueous extracts of 18 Lamiaceae plants from Serbia was evaluated. The total coumarins and triterpenes were detected spectrophotometrically, while a microdilution assay measured their effects on bacterial growth. Additionally, the impact of these extracts was assessed on *Pseudomonas aeruginosa* PAO1 adhesion and invasion in human fibroblasts and biofilm formation and degradation. The alcoholic extracts had the highest phytochemical content, with *Teucrium montanum* and *Lavandula angustifolia* being the richest in coumarins and triterpenes, respectively. Gram-positive bacteria, particularly *Bacillus subtilis*, were more susceptible to the extracts. *Hyssopus officinalis* ethanolic and *Sideritis scardica* methanolic extracts inhibited bacterial growth the most efficiently. Although the extracts did not inhibit bacterial adhesion, most ethanolic extracts significantly reduced bacterial invasion. *Origanum vulgare* and *H. officinalis* ethanolic extracts significantly inhibited biofilm formation, while *Teucrium chamaedrys* extract was the most active in biofilm degradation. This study significantly contributes to the literature by examining the antibacterial activity of Lamiaceae extracts, addressing major literature gaps, and underscoring their antibacterial potential, particularly *Satureja montana* and *O. vulgare* ethanolic extracts, linking their efficacy to coumarins and triterpenes.

## 1. Introduction

Infectious diseases are considered responsible for significant health loss, annually claiming the lives of millions of people worldwide (13.7 million infection-related deaths, of which 7.7 million deaths were caused by 33 bacterial pathogens in 2019) [1]. With the clinical use of antibiotics, the development of bacterial infections has significantly slowed down [2]. However, in recent decades, and especially in the post-COVID-19 era, their efficacy has diminished due to both their overuse and misuse, as well as the ability of bacteria to develop mechanisms of tolerance or resistance to antibiotics [3,4,5]. It is estimated that in developed countries, over 60% of human bacterial infections are caused by bacterial biofilms, while the removal of biofilms represents a serious health, social, and economic difficulty that people face nowadays [5,6]. The reason for this is that bacterial cells within the biofilm are more resistant to antimicrobial agents than planktonic bacteria, which is a major challenge in the therapy of certain infections [7]. Particularly, due to their antibiotic resistance, multi-resistant strains, including *Enterococcus faecium*, *Staphylococcus aureus*, *Klebsiella pneumoniae*, *Acinetobacter baumannii*, *Pseudomonas aeruginosa*, and *Enterobacter* spp., collectively referred to as “ESKAPE” pathogens, pose a serious and significant obstacle in treating nosocomial infections [5,8].

Persisters, a subset of bacterial cells, exhibit transient tolerance to antibiotics. They typically exhibit slow or halted growth in the presence of antibiotics but can resume growth after exposure to lethal stress. This formation of persistent cells introduces phenotypic heterogeneity within the bacterial population, a critical mechanism for adapting to environmental changes [9]. Moreover, since biofilms can form on abiotic surfaces, there is an increased interest in finding ways to suppress them, especially from medical devices and equipment such as catheters. Additionally, the release of microorganisms from biofilms can often cause life-threatening infections [5,8,10,11]. Commonly used antibiotics impose increased selective pressure for multi-drug resistance and often lack selectivity, affecting both pathogenic and commensal bacteria’s biochemical and physiological functions [5]. Hence, there is an urgent demand for safe and effective alternatives that can either substitute or complement existing antibiotics while remaining unchallenged by bacterial resistance [4,5]. Given that more than 70 years ago, before the advent of antibiotics, 22% of medicinal drugs originated from herbs, nowadays there is a renewed interest in exploring the antibacterial activity of plants, their essential oils, extracts, and secondary metabolites [2]. It has been reported that phytoconstituents such as phenolics, coumarins, terpenoids, and alkaloids have various mechanisms of action, among which are the disturbance of the integrity of the bacterial membrane; inactivation of bacterial proteins, adhesins, and other enzymes; blocking of cell-to-cell signalization; and also the inhibition of biofilm formation and the promotion of biofilm degradation [9,12,13].

Essential oils from Lamiaceae representatives have previously been studied for their antibacterial potential. It has been demonstrated that they can, independently or in combination with existing antibiotic drugs, inhibit bacterial cell growth, diminish the virulence of highly resistant strains such as MRSA (Methicillin-resistant *Staphylococcus aureus*), and both hinder the formation and induce the degradation of pre-existing biofilms in specific bacteria [14,15,16,17,18,19,20,21]. However, to this day, there is a limited amount of research available on the antibacterial mechanisms of extracts derived from these plants. Hence, this study aimed to determine the antibacterial potential of methanolic, ethanolic, and aqueous extracts derived from 18 aromatic and medicinal Lamiaceae plant species traditionally used in Serbia by employing a multi-tier study design. Firstly, the total contents of coumarins and triterpenes were determined. Afterwards, the lowest concentration of an extract that inhibited the bacterial growth (minimum inhibitory concentration—MIC) was assessed on four Gram-positive (*Bacillus subtilis*, *Enterococcus faecalis*, *Listeria innocua*, and *Staphylococcus aureus*) and three Gram-negative bacteria (*Escherichia coli*, *Pseudomonas aeruginosa*, and *Salmonella enterica* subsp. *enterica* serovar Typhimurium—*Salmonella typhimurium*). Additionally, the effects of the ethanolic extracts on adhesion and invasion of *P. aeruginosa* PAO1 during lung fibroblast infection (MRC-5 cells) were examined. *P. aeruginosa* PAO1 was chosen as a model in this study, as it is the most commonly used strain for research, being highly resistant to existing antibiotics and disinfectants and particularly responsible for severe hospital infections, which makes its removal from hospital environments and the treatment of infected patients particularly difficult [22,23]. Subsequently, the efficacy of selected ethanolic extracts on bacterial biofilm was studied in terms of their impact on *P. aeruginosa* PAO1 biofilm formation and degradation. The results were standardized using the Integrated Biomarker Response (IBR) technique to reveal the extract with the highest antibacterial potential.

## 2. Results

### 2.1. Yield of Extracts

Extracts of the studied plants were obtained by using three solvents, methanol, ethanol, and water, to extract components of different polarities. The percentage yield of the extracts varied with the solvent used for extraction (Table 1).

Aqueous extracts of *H. officinalis*, *L. angustifolia*, *M. officinalis*, *O. majorana*, *R. officinalis*, *S. montana*, *T. chamaedrys*, *Th. serpyllum*, and *Th. vulgaris* had a higher yield than their methanolic and ethanolic extracts. Methanolic extracts of *G. hederacea*, *M. vulgare*, *M. piperita*, *O. basilicum*, *O. vulgare*, *S. scardica*, and *T. montanum* had higher yields compared to other solvents. Ethanolic extracts of *L. cardiaca* and *S. officinalis* had higher yields compared to their methanolic and aqueous extracts. *M. vulgare* ethanolic extract had the highest yield of all the tested plants (34.1%), while the lowest yield was obtained for the aqueous extract of *G. hederacea* (9.07%) (Table 1).

### 2.2. Total Coumarin and Triterpene Contents in Lamiaceae Extracts

Since the total coumarin and triterpene contents were not detected at 100 and 250 μg/mL of extracts, only the results using the highest tested concentration of extracts (500 μg/mL) are shown in Figure 1 and Figure 2.

The majority of the examined members from the Lamiaceae family have significant amounts of total coumarins. The highest level of coumarins was detected in alcoholic extracts—methanolic <1 to 161.22 mg CE/g, ethanolic 11.22 to 126.53 mg EC/g—while aqueous extracts had the lowest amount of coumarins (<1–97.96 mg CE/g) (Figure 1). The methanolic extract of *T. montanum* contained the highest amount of total coumarins (161.22 mg CE/g), while among the ethanolic extracts, the *G. hederacea*, *H. officinalis,* and *M. vulgare* ones had significantly more of these compounds in comparison to the methanolic and aqueous extracts. The aqueous extracts of *L. angustifolia* and *O. vulgare* contained significantly more coumarins compared to the extracts prepared using the other two solvents. It should be noted that in certain extracts, coumarins were not detected, such as the methanolic extracts of *H. officinalis*, *L. angustifolia*, and *M. vulgare*, as well as aqueous extracts of *G. hederacea*, *H. officinalis*, and *M. vulgare*.

As was the case with total coumarins, in certain extracts, triterpenes were not detected even in the highest tested concentration. Overall, the highest amounts of triterpenes were detected in methanolic extracts (range from <1 to 210.42 mg UAE/g), while the aqueous extracts contained the lowest amounts of triterpenes (<1–52.12 mg UAE/g). All the ethanolic extracts had a measurable amount of triterpenes in the range of 1.61–109.4 mg UAE/g. The most triterpenes were found in *L. angustifolia* extracts, especially in the methanolic one (210.42 mg UAE/g), which contained almost two times more triterpenes than the ethanolic extract and four times more triterpenes than the aqueous extract. Finally, the lowest content of triterpenes was found in *O. basilicum* extracts (Figure 2).

### 2.3. Inhibition of Bacterial Growth by Lamiaceae Extracts

The MIC and MBC of methanolic and ethanolic extracts of the examined Lamiaceae representatives are presented in Table 2. The results for the aqueous extracts are omitted due to their lack of antibacterial activity. Additionally, results for the inhibition of *E. coli* and *S. typhimurium* growth are not provided in Table 2, as their MICs and MBCs exceeded the highest tested concentration of extracts, 1000 µg/mL. In general, the antibacterial effect of the extracts was more pronounced against Gram-positive compared to Gram-negative bacteria, except for *E. faecalis*. Of all the tested bacteria, *B. subtilis* showed the highest sensitivity; *H. officinalis* ethanolic extract and *S. scardica* methanolic extract inhibited *B. subtilis* at the same MIC value, 31.25 µg/mL, while the MIC of *S. officinalis* ethanolic extract was 62.5 µg/mL. In contrast to *E. coli* and *S. typhimurium*, where the MIC for all tested extracts was higher than 1000 µg/mL, the MIC values of the ethanolic extracts for *P. aeruginosa* were in the range of 250–500 µg/mL (Table 2). The positive control, streptomycin, showed the strongest antibacterial activity (*B. subtilis* 3.125 µg/mL, *E. coli* 12.5 µg/mL, and *S. typhimurium* 3.125 µg/mL), while the negative controls, ethanol and methanol, in the tested concentrations, did not affect the growth of the tested bacteria.

### 2.4. Inhibition of Adhesion and Invasion of P. aeruginosa PAO1 on Human Fibroblasts by Lamiaceae Extracts

In the adhesion assay, the tested ethanolic extracts did not inhibit the adhesion of *P. aeruginosa* PAO1 to the MRC-5 cells.

On the other hand, all the tested ethanolic extracts, except *H. officinalis* and *M. officinalis*, significantly reduced the number of bacteria capable of invading MRC-5 cells (Figure 3).

### 2.5. Inhibition of P. aeruginosa PAO1 Biofilm Formation and Degradation by Lamiaceae Extracts

*P. aeruginosa* PAO1 was used to monitor the inhibitory effect of ethanolic extracts on the formation and degradation of biofilm. The selection of extracts for these assays was based on the obtained MIC values for *P. aeruginosa* ATCC 15442. All extracts with an MIC of 250 µg/mL (Table 2) were tested, except for *S. scardica*, due to its low solubility.

In the biofilm formation inhibition assay, ethanolic extracts were tested in the range from 156 to 1250 µg/mL (⅛MIC–MIC), and the results are presented in Figure 4. All tested extracts showed an inhibitory effect on biofilm formation. The extracts of *O. vulgare* and *H. officinalis* demonstrated the highest overall level of biofilm formation inhibition, with significant inhibition rates of 45.43% and 49.1%, respectively, both at a concentration of 625 µg/mL. Moreover, *M. officinalis*, *M. piperita*, *R. officinalis*, *S. montana*, *T. chamaedrys*, *T. montanum*, *Th. serpyllum*, and *Th. vulgaris* extracts at certain concentrations also exhibited significant inhibition of the formation of *P. aeruginosa* PAO1 biofilm. It is important to note that, except for *M. piperita*, *O. basilicum*, and *T. montanum*, the highest concentrations of extracts did not result in the strongest inhibition. This indicates that there was no concentration-dependent inhibition observed. The negative control, ethanol (tested in the range of 1.092–8.75%), did not affect the viability of bacteria during biofilm formation. Hence, the statistical analysis was performed to assess the significance of differences between the extracts and ethanol to determine the extent to which the extracts themselves contribute to biofilm inhibition. Also, streptomycin (tested in the range of 0.781 to 6.25 µg/mL) did not cause a reduction in bacterial viability (Figure 4).

The effects of the selected extracts on the *P. aeruginosa* PAO1 biofilm degradation are shown in Figure 5. The extracts were tested at twice the concentration used in the inhibition of biofilm when performing the biofilm formation assay (from 312 to 2500 µg/mL, ⅟4MIC–2 × MIC), since the degradation of already-formed biofilm requires agents of stronger activity compared to the ones used for the inhibition of its formation. Among the tested extracts, *T. chamaedrys* extract displayed the best activity, degrading 42.26% of the pre-existing biofilm at 625 µg/mL. Similar to the assessment of the Lamiaceae extracts’ effects on biofilm formation, several extracts showed statistically significant differences in biofilm degradation compared to ethanol, namely, *H. officinalis*, *M. piperita*, *O. basilicum*, *O. majorana*, *O. vulgare*, *S. montana*, *T. montanum*, *Th. Serpyllum,* and *Th. vulgaris* extracts, which at lower concentrations showed a significant potential for biofilm degradation. Ethanol was tested in the range of 2.184–17.5% and streptomycin in the range of 1.562–12.5 µg/mL. Ethanol degraded 22.48% of the pre-existing biofilm at the highest tested concentration, and streptomycin acted in a dose-dependent manner, leaving only 28.3% of bacteria viable at the highest tested concentration (Figure 5).

### 2.6. Correlation Analysis across Assays: Phytochemical Content, Bacterial Growth, Invasion, Biofilm Formation, and Degradation

The correlations between the total coumarin and triterpene contents of the extracts and the results of the antibacterial assays, as well as the inter-correlation of these results, were defined by Pearson correlation coefficients (r). The calculated individual r values in Table 3 depict only the results of ethanolic extracts, either because only these were examined or due to their higher antibacterial activity compared to that of the methanolic and aqueous extracts.

The coumarin content correlated moderately with the inhibition of *B. subtilis*, *E. faecalis*, and *P. aeruginosa* growth, as well as with the degradation of pre-existing *P. aeruginosa* PAO1 biofilm. However, only the correlations between coumarin content and the inhibition of *B. subtilis* and *P. aeruginosa* growth were statistically significant (Table 3).

The results of the MIC assay obtained for the different bacteria exhibited variations, with a weak correlation observed between the MIC assay results of Gram-positive and Gram-negative bacteria. Notably, a significant, strong correlation was identified only between the inhibition of *B. subtilis* and *S. aureus* growth. Furthermore, the correlation found between the inhibition of *E. faecalis* and *L. innocua* was identified as significant and moderate. Moderate correlations were also found among the inhibition of *S. aureus*, *E. faecalis,* and *L. innocua* growth (Table 3).

The findings from the *P. aeruginosa* PAO1 invasion assay exhibited moderate correlation with the inhibition of *L. innocua* growth, while only their moderate correlation with the inhibition of *P. aeruginosa* PAO1 biofilm formation was identified as statistically significant (Table 3).

The assays employed to assess the extracts’ capacity to inhibit the formation and promote the degradation of *P. aeruginosa* PAO1 biofilm exhibited a moderate inter-correlation. Additionally, a moderate correlation was identified between the ability of the extracts to inhibit the formation of *P. aeruginosa* PAO1 biofilm and the growth of *E. faecalis*. The correlation between the ability of the extracts to inhibit the formation of *P. aeruginosa* PAO1 biofilm and the growth of *P. aeruginosa* was identified as moderately and statistically significant. On the other hand, the ability of extracts to degrade *P. aeruginosa* PAO1 biofilm had a moderate and significant correlation with the growth inhibition of *B. subtilis* and *S. aureus* (Table 3).

### 2.7. Integrated Biomarker Response Analysis

Finally, the acquired results underwent standardization using the Integrated Biomarker Response (IBR) analysis technique. The IBR analysis included the following:standardization of the MIC assay results for *B. subtilis*, *E. faecalis*, *L. innocua*, *S. aureus*, and *P. aeruginosa* ATCC 15442;*P. aeruginosa* PAO1 invasion assay in lung fibroblast infection;inhibition of *P. aeruginosa* PAO1 biofilm formation;inhibition of *P. aeruginosa* PAO1 biofilm degradation.

The IBR analysis of antibacterial activity included the standardization and graphical presentation of results obtained for the ethanolic extracts of the following Lamiaceae representatives: *H. officinalis*, *M. officinalis*, *M.* piperita, *O. basilicum*, *O. majorana*, *O. vulgare*, *R. officinalis*, *S. montana*, *T. chamaedrys*, *T. montanum*, *Th. serpyllum*, and *Th. vulgaris* (Figure 6 and Figure 7). The MIC assay on *E. coli* and *S. typhimurium* and the *P. aeruginosa* PAO1 adhesion assay in lung fibroblast infection were excluded from this analysis as none of the ethanolic extracts exhibited activity in these assays.

Results from the IBR analysis revealed that the ethanolic extract of *S. montana* displayed the highest antibacterial potential, followed by *O. vulgare* and *T. chamaedrys* extracts. In contrast, the ethanolic extract of *T. montanum* demonstrated the weakest antibacterial potential (Figure 6 and Figure 7).

## 3. Discussion

Plants with antibacterial potential are increasingly being researched on, as it was demonstrated that their metabolites, including phenolic compounds and terpenes, have better bioavailability compared to synthetically derived drugs. Current studies show that plant extracts, essential oils, and their isolated phytoconstituents affect the growth of both Gram-positive and Gram-negative bacteria, influencing adhesion and invasion during infections in various cells and tissues, as well as the dynamics of bacterial biofilm formation [25].

While many literature references align with our results of the yield of extraction [26,27,28,29], certain studies indicate variations in the extraction yield for the examined extracts [30,31,32,33]. Research has demonstrated that differences in extraction yield values are significantly influenced by factors such as the chosen extraction protocol, extraction time, solvent polarity, and, most importantly, the plant species [34,35]. Additionally, one of the most critical factors is the temperature applied in the extraction procedure, specifically the solvent temperature. Kivilompolo and Hyötyläinen [36] validated that a higher temperature enhances the extraction yield. Hence, the higher yield observed in our study for aqueous extracts can be attributed to the use of boiling, rather than cold, distilled water in the extraction procedure.

Given the low extraction yields for both methanol and ethanol in our study, and the use of ethanol in developing various plant-based products, optimizing the extraction protocol is essential for future applications. Enhancing extraction parameters and incorporating advanced techniques, such as ultrasound-assisted extraction and binary solvent systems, can significantly improve the extraction yield. Combining methanol or ethanol with water in optimized ratios enhances solvent polarity and improves the solubility and extraction efficiency of phytochemicals. Even small amounts of water have been reported to significantly enhance the extraction outcome [34,35]. Moreover, ultrasound-assisted extraction disrupts cell walls and facilitates the release of bioactive compounds [37], while increasing the solid/solvent ratio prevents saturation of the extraction medium, resulting in improved extraction yield [38]. Reducing solvent volumes makes the process more sustainable and cost-effective, especially with environmentally friendly solvents like ethanol. Comparative studies of methanol, ethanol, and their binary mixtures can identify the most efficient solvent systems for specific bioactive compounds. By integrating these advanced techniques and optimizing parameters, future research can significantly improve extraction yields, enhancing practical applications.

Coumarins (2H-1-benzopyran-2-ones) are structurally diverse phenolic compounds that exhibit a myriad of biological activities, such as antioxidant, antibacterial, antifungal, antiviral, cytotoxic, and antitumor activities [39]. In our study, among the methanolic extracts, the *T. montanum* one had the highest coumarin content. Among the ethanolic extracts, the *M. piperita* one was the richest in coumarins, while among the aqueous extracts, the *S. officinalis* one had the highest coumarin content. Our review of the literature indicated that the analysis of coumarin content in the 18 Lamiaceae representatives examined in this study has not been frequently conducted up to the present day. Nevertheless, Patil et al. [40] demonstrated that the aqueous extract of *M. piperita* from India had a higher coumarin content compared to the ethanolic extract, in contrast to our findings, which revealed a higher coumarin content specifically in the ethanolic extract. Moreover, Mahdi et al. [41] demonstrated the presence of coumarins and terpenes in the ethanolic extract of *S. officinalis*. Although the specific composition of coumarins in Lamiaceae representatives is infrequently analyzed; it was documented that *O. basilicum* and *S. officinalis* contain esculetin (6,7-dihydroxycoumarin), with *O. basilicum* also containing esculin (6,7-dihydroxycoumarin-6-glucoside), while *L. angustifolia* possesses coumarin, herniairin (7-methoxycoumarin), santonin, and umbelliferone, along with lavnadupyrone A and B [42,43,44]. Despite previous reports, the extracts of *L. angustifolia* examined in our studies did not contain a significant amount of coumarins.

Triterpenes are a class of secondary metabolites belonging to the terpene family, characterized by a structural motif composed of six isoprene units [45]. Triterpenes are commonly present in plants, and they contribute to the pharmacological and therapeutic potential of these plants, playing a crucial role in the development of new bioactive products [46,47]. While the antibacterial activity of terpenes from essential oils is well-established [12], their exploration and characterization in Lamiaceae family plant extracts have been relatively limited. This is partly due to the unsaturated carbon bonds in terpenes, making them susceptible to reactions with atmospheric oxidants such as OH^−^, O_3_, and NO_3_. As a result, terpenes have a short lifespan (typically ranging from a few minutes to a few hours), making their identification and accurate quantification in plant material challenging [48]. In our study, similar to coumarins, the concentration of triterpenes in the extracts showed statistically significant variations depending on the extraction solvent. *L. angustifolia* extracts, particularly the methanolic one, exhibited the highest triterpene content. *Lavandula* species are renowned for having a high terpene content, making the identification of *L. angustifolia* extracts as the richest in triterpenes among all tested extracts unsurprising. Héral et al. [44] reported in their study that plants from the *Lavandula* genus contain over 30 different triterpenes, including tetracyclic, pentacyclic, and steroid derivatives. Notable triterpenes include betulin, 2α-hydroxyursolic, 2,3-hydroxytormentic, 3-epiursolic, betulinic, micromeric, oleanolic, and ursolic acids, as well as α-amyrin, β-amyrin, and uvaol. Moreover, previous research indicates that extracts of *M. vulgare* contain high amounts of triterpenes [32], a finding not consistent with our results. In our study, *O. basilicum* extracts displayed the lowest amount of triterpenes. Although the examination of total triterpene content in *O. basilicum* extracts was not conducted previously, it was found that its ethyl acetate extract exhibited a high content of total terpenes [48]. The existing literature data diverge from our study results, likely due to the authors quantifying total terpenes, which encompass various classes such as mono-, di-, and triterpenes. Additionally, it is important to note that the choice of solvent for plant material extraction can significantly influence the content of these secondary metabolites in the sample. To the best of our knowledge, there are no previous reports on the total triterpene content for the rest of the Lamiaceae representatives that were investigated in our study.

In our investigation, the MIC assay was employed to assess the antibacterial activity of extracts from 18 Lamiaceae representatives. Additionally, all extracts underwent further examination in the *P. aeruginosa* PAO1 adhesion and invasion assays. Moreover, selected ethanolic extracts were subjected to assays evaluating their ability to inhibit the formation and degradation of *P. aeruginosa* PAO1 biofilm. The Pearson’s correlation results indicated a probable association between coumarins and triterpenes with the displayed antibacterial potential of Lamiaceae extracts. Particularly, coumarins showed a significant correlation with the growth inhibition of *B. subtilis* and *P. aeruginosa* ATCC 15442, as well as with the degradation of *P. aeruginosa* PAO1 ATCC 15692 biofilm. Indeed, previous studies indicate that the multifaceted nature of coumarins allows them to interfere with various bacterial processes, which makes plants containing them potential candidates for combating bacterial infections [49].

In the MIC assay, the results revealed that the ethanolic extracts exhibited the highest activity. Unlike their alcoholic counterparts, aqueous extracts displayed no activity. The lack of antibacterial activity observed in aqueous extracts of Lamiaceae representatives can be attributed to the testing of relatively low extract concentrations (the highest tested concentration was 1000 μg/mL), as well as the insufficient solubilization of hydrophobic bioactive compounds from these plants [34,35]. These hydrophobic compounds, which often contribute significantly to the antibacterial properties, may not be sufficiently extracted in water-based solutions, leading to lower efficacy against bacterial growth compared to extracts obtained with other organic solvents. Additionally, antibacterial activity is frequently associated with lipophilic phytocomponents with physicochemical properties that influence their ability to diffuse and dissolve within bacterial membranes, leading to various antibacterial effects. Nonetheless, they are more effectively extracted in non-polar solvents such as ethanol or methanol, as opposed to water [50].

Previous studies have explored the impact of extracts of Lamiaceae representatives on bacterial growth activity [40,51,52,53,54,55,56,57,58,59]; however, our literature review revealed that current research mainly focuses on the antibacterial properties of essential oils derived from Lamiaceae plants [12,50,60,61,62]. In our study, *B. subtilis* showed the highest susceptibility to the effects of Lamiaceae extracts, with *H. officinalis* and *S. officinalis* ethanolic and *S. scardica* methanolic extracts demonstrating the most potent growth-inhibitory activity. Our literature survey uncovers that Lamiaceae extracts typically exhibit inhibition of bacterial growth at concentrations surpassing those examined in our study, reaching as high as 40 mg/mL, which was specifically found for *M. officinalis* ethanolic extract and the inhibition of *E. coli* and *P. aeruginosa* growth [63]. Despite the previous study, it is noteworthy that the *S. officinalis* extracts investigated by Mocan et al. [64] exhibited high inhibition against Gram-negative bacteria growth (*E. coli* at 45 μg/mL and *P. aeruginosa* and *S. typhimurium* at 90 μg/mL) in contrast to our results. Additionally, the results of our MIC assay for the different bacteria exhibited variations, indicating a weak correlation between the MIC assay results for Gram-positive and Gram-negative bacteria. This outcome was expected due to the presence of an outer membrane with a hydrophilic polysaccharide chain in Gram-negative bacteria, acting as a hydrophobic barrier against many plant secondary metabolites [25]. Therefore, similar to essential oils [62], plant extracts may encounter difficulties in efficiently targeting the phospholipid layers of Gram-negative bacterial cells, potentially compromising their permeability and structural integrity.

The infection of invasive bacteria, exemplified by *P. aeruginosa*, is marked by three stages: (i) adhesion and colonization, (ii) local infection through tissue penetration and internalization, and (iii) dissemination through the bloodstream [65]. Initial tissue penetration stages (extracellular matrix protein and tight junction cleavage) and host cell invasion are pivotal for bacterial survival and infection initiation. In *P. aeruginosa* PAO1, cell invasion, facilitated by the secretome (comprising toxins, proteases, lipases, and lysines), occurs independently of lipopolysaccharide production or cytotoxicity. Notably, these bacteria exhibit a higher binding affinity for inflamed or compromised cells, emphasizing the importance of employing natural antioxidants and anti-inflammatory agents, such as plant extracts, particularly in pathological conditions [9,66,67].

While our study indicated that methanolic, ethanolic, and aqueous extracts of Lamiaceae species did not affect *P. aeruginosa* PAO1 adhesion to MRC-5 cells, earlier investigations [15,67,68,69,70] demonstrated inhibition of bacterial adhesion to human cells for extracts of *L. cardiaca*, *M. vulgare*, *M. piperita*, *R. officinalis*, *S. montana*, and *Th. vulgaris*.

Furthermore, our findings indicated that all ethanolic extracts of the evaluated Lamiaceae representatives, except for *H. officinalis* and *M. officinalis*, significantly reduced the number of bacteria capable of invading MRC-5 cells. It is important to emphasize that there are no previous reports on the bacterial invasion of human cells by these plants, except for the study of Šimunović et al. [70], who demonstrated that the *S. montana* ethanolic extract (62.5 μg/mL) inhibited the invasion of *Campylobacter jejuni* 11168 into INT407 epithelial cells by up to 81%, a percentage higher than the one observed in our results.

Additionally, *P. aeruginosa*, a prominent biofilm former, serves as a valuable model for studying the dynamics of biofilm formation and degradation. Gaining insight into biofilm composition, structure, and the molecular mechanisms contributing to antibacterial tolerance is essential for developing strategies to not only manage and prevent but also eradicate biofilm-associated infections. However, the treatment of *P. aeruginosa* displays distinctive challenges in utilizing most of the available antibiotics since it showcases multi-drug resistance mechanisms [71]. Research on the dynamics of *P. aeruginosa* biofilm, especially its degradation, has been scarce for Lamiaceae plant extracts. In addition to studies that have researched the dynamics of *P. aeruginosa* biofilm formation and/or degradation [54,72,73,74,75,76], there are several studies focused on the effect of these extracts on the biofilms of *Bacillus cereus*, *C. jejuni*, *S. aureus*, *E. coli*, *Acinetobacter baumannii*, *Klebsiella pneumoniae*, and MRSA [15,54,68,73,74,77,78]. Our findings indicate that ethanolic extracts of *O. vulgare* and *H. officinalis* significantly inhibited *P. aeruginosa* PAO1 biofilm formation by up to 50%. Additionally, at certain concentrations, extracts from *M. officinalis*, *M. piperita*, *R. officinalis*, *S. montana*, *T. chamaedrys*, *T. montanum*, *Th. serpyllum*, and *Th. vulgaris* significantly inhibited the formation of *P. aeruginosa* PAO1 biofilm. Moreover, *T. chamaedrys* showed the best activity, degrading 42.26% of pre-existing biofilm at 625 µg/mL. Additionally, *H. officinalis*, *M. piperita*, *O. basilicum*, *O. majorana*, *O. vulgare*, *S. montana*, *T. montanum*, *Th. serpyllum*, and *Th. vulgaris* extracts demonstrated significant biofilm degradation at lower concentrations. Interestingly, the extracts that exhibited the highest inhibition of biofilm formation were not as effective in degrading a pre-existing biofilm. However, this outcome is anticipated as plant extracts and their phytoconstituents may employ various mechanisms for their anti-biofilm activities: (i) plant extracts, containing compounds like phenolics, can modify the microenvironment around bacterial cells and disrupt bacterial communication systems, specifically quorum sensing, crucial for biofilm formation, thereby impeding the coordination of bacterial cells in biofilm development [79,80,81]; (ii) plant extracts and phenolic compounds, specifically flavonoids, directly inhibit the growth of *P. aeruginosa* cells and their adherence to surfaces, preventing the initial stages of biofilm formation [82]; and (iii) phytoconstituents may disrupt the biofilm matrix by enzymatically breaking down the extracellular polymeric substances, making the biofilm structure more susceptible to degradation [10]. The precise mechanisms rely on the composition of the plant extract and their bioactive compounds. Altogether, it was proven that various plant species and their extracts may exhibit unique properties that contribute to their effectiveness in both inhibiting biofilm formation and degrading existing biofilms [83].

Last but not least, our IBR analysis identified the ethanolic extracts of *S. montana* and *O. vulgare* as highly promising antibacterial agents. This observation is noteworthy as prior studies predominantly emphasized the antibacterial activity of their essential oils [50,61,84]. Our findings further reveal the promising antibacterial efficacy of their extracts, shedding light on their potential as a powerful natural defense against various aspects of bacterial infections.

## 4. Materials and Methods

### 4.1. Plant Material

The experimental plant material, commercially available and sourced from the Institute for Medicinal Plant Research “Dr. Josif Pančić” (IMPR) in Belgrade, Serbia, comprises 18 medicinal, aromatic, and spice Lamiaceae species from Serbia. Collected during the spring of 2018, vouchers for each plant species used in this study have been cataloged in the IMPR’s herbarium (Table 4).

### 4.2. Chemicals and Reagents

Ethanol, glacial acetic acid, hydrochloric acid, and methanol were bought from Zorka Pharma, Šabac, Serbia. Rifampicin was obtained from Hemofarm, Belgrade, Serbia; lead acetate trihydrate was obtained from Superlab, Belgrade, Serbia; while streptomycin and gentamicin were obtained from Galenika, Belgrade, Serbia. Coumarin, crystal violet, DMSO (dimethyl sulfoxide), glucose, magnesium sulfate, sodium chloride, Triton X-100, trypan blue, tryptone, ursolic acid, and vanillin were purchased from Merck, Rahway, NJ, USA. DMEM 5523 (Dulbecco’s Modified Eagle Medium), FBS (Fetal Bovine Serum), and PBS (Phosphate-Buffered Saline) were purchased from Gibco, Invitrogen, Waltham, MA, USA, while perchloric acid was obtained from VWR, Radnor, PA, USA. Agar, Brain-HeartInfusion (BHI), and yeast extract were obtained from Lab M Ltd., Neogen, Heywood, UK. Resazurin sodium salt (>90% (LC)) was bought from SERVA Electrophoresis GmbH, Heidelberg, Germany, while penicillin–streptomycin solution was purchased from PAA Laboratories GmbH, Pasching, Austria. Muller–Hinton Broth (Himedia, MHB, Maharashtra, India) was obtained from Biomedics, Málaga, Spain.

### 4.3. Preparation of Lamiaceae Extracts

Pre-ground plant material (10 g) underwent extraction using the classic maceration method (10% *w*/*v*, 24 h at 25 °C) [33]. Three solvents—70% methanol, 70% ethanol, and boiling distilled water (100 °C)—were employed for the extraction process. Ultrasonic treatment was applied for a total of two hours (one hour before and one hour after maceration, 30 °C). The resulting mixture underwent double filtration with Whatman No. 1 filter paper, followed by the removal of excess solvent using a Büchi rotavapor R-114 evaporator under reduced pressure. The resulting crude extracts were stored in glass vials at 4 °C until subsequent experimentation.

#### Yield of Extracts

The dry extract’s mass was determined for each sample, and the yield was calculated using the following equation:Yield (%) = [(m_1_ − m_0_)/M] × 100,(1)
where m_0_ represents the mass of the empty vial; m_1_ is the mass of the vial with the extract; (m_1_ − m_0_) is the mass of the dry extract; and M is the mass of the dry plant material used for extraction. The results are expressed as percentages.

### 4.4. Total Coumarin Content

The method for determining the total coumarin content followed the procedure by de Amorim et al. [85] with some modifications, conducted in 96-well microtiter plates. An amount of 2 μL of extract at concentrations of 100, 250, and 500 μg/mL were dispensed into the wells, followed by the addition of 8 μL distilled water and 2 μL lead acetate solution (5% w/v). Additionally, 28 μL distilled water and 160 μL of 0.1 M hydrochloric acid were added to each well. A blank was prepared with all components, substituting the sample with an appropriate solvent (70% methanol, 70% ethanol, or distilled water). The microtiter plate with the reaction mixtures and the blank was incubated for 30 min at room temperature, and absorbances were measured at 320 nm using a Multiskan Sky Thermo Scientific microtiter plate reader, Vantaa, Finland.

To generate the calibration curve, coumarin (C) dissolved in methanol was utilized in concentrations ranging from 5 to 1000 μg/mL instead of the extract. All the samples were tested in triplicate. The total coumarin content in the samples was calculated using the calibration curve equation (y = 0.2440x + 0.0093; R^2^ = 0.9983) and expressed in coumarin equivalents as mg CE/g of dry extract. The results are presented as the mean of three replicates ± standard error.

### 4.5. Total Triterpene Content

The method for determining total triterpene content followed the procedure by Chang et al. [86], with adjustments, using 96-well microtiter plates. In brief, 10 μL of extract at concentrations of 100, 250, and 500 μg/mL were dispensed into the well, followed by the addition of 15 μL vanillin–glacial acid solution (5% *w*/*v*) and 50 μL perchloric acid. The microtiter plate with the reaction mixtures was incubated at 60 °C for 45 min, then cooled to 25 °C on ice, and afterward, 225 μL of glacial acetic acid was added to each well. A blank, containing all the components except the sample, substituted with an appropriate solvent (70% methanol, 70% ethanol, or distilled water), was prepared. Absorbances were measured at 548 nm using a Multiskan Sky Thermo Scientific microtiter plate reader, Finland. For the calibration curve, ursolic acid (UA) dissolved in 100% methanol (concentration of 5 to 1000 μg/mL) was used instead of the extract. Each sample was tested in triplicate.

The total triterpene content of the samples was calculated from the calibration curve equation (y = 0.0008x + 0.0082, R^2^ = 0.995) and expressed in UA equivalents as mg UAE/g of dry extract. The results are presented as the mean of three replicates ± standard error.

### 4.6. Preparation of Bacterial Strains

The Gram-positive and Gram-negative bacterial strains employed in the experimental work are part of the ATCC collection (Table 5).

A colony of the respective bacteria was transferred to a test tube containing 5 mL of MHB medium, pre-warmed to 37 °C, and allowed to cultivate overnight at 37 °C.

To ascertain the minimum inhibitory concentration (MIC) of the extracts, the bacteria listed in Table 5 were cultured to the exponential growth phase, providing the required bacterial density per milliliter for each strain [87]. Subsequently, a dilution of 0.01 M magnesium sulfate was made for each strain to achieve a final bacterial concentration in the inoculum of 10^5^ CFU (colony forming unit)/mL which was used to determine the MIC.

The impact on both biofilm formation and the degradation of existing biofilm was examined in *P. aeruginosa* PAO1 ATCC 15692. The optical density (OD_600_) was adjusted to 0.4 using an MHB medium, resulting in a bacterial concentration of 10^8^ CFU/mL.

### 4.7. Cell Culture Preparation for Assessing the Impact of Lamiaceae Extracts on Adhesion and Invasion of P. aeruginosa PAO1 ATCC 15692

MRC-5 cells (ECACC no. 84101801) were cultivated in 75 cm^2^ flasks, and 1 × 10^5^ cells were subsequently transferred to 12-well microtiter plates. The cells were then incubated in DMEM complete (with fetal bovine serum) nutrient medium at 37 °C with 5% CO_2_ for 48 h to achieve a confluent monolayer.

### 4.8. Microdilution Assay

The MICs of the methanolic, ethanolic, and aqueous extracts were determined by the microdilution method in 96-well plates, using a series of double dilutions. Resazurin (final concentration 675 μg/mL) served as a bacterial growth indicator. Sample stocks containing 10 mg/mL extracts were used to prepare two-fold gradient dilutions in 96-well plates with the highest tested concentration of 1000 μg/mL, 200 µL total volume. Each well contained 20 µL of pre-prepared bacterial inoculum (10^5^ CFU/mL). Resazurin (22 µL) was added the same day.

Microtiter plates were incubated at 37 °C for 24 h. To assess the bacteriostatic or bactericidal effect, solid nutrient media (MHA) screening was conducted, followed by a 24 h incubation at 37 °C. Bacterial growth presence or absence determined the substance’s bacteriostatic or bactericidal effect, with the lowest concentration at which there was no bacterial growth being referred to as minimum bactericidal concentration (MBC) and the concentration at which bacterial growth occurred being referred to as minimum bacteriostatic concentration (MIC). The following controls were included in the experiment: (i) medium sterility; (ii) solvents (for methanol and ethanol); (iii) bacterial growth (negative control); and (iv) relevant antibiotics (positive controls).

### 4.9. Assessment of Lamiaceae Extracts Impact on P. aeruginosa PAO1 Adhesion and Invasion in Lung Fibroblast Cell Infection

#### 4.9.1. Adhesion Assay

Confluent MRC-5 cell monolayers in a 24-well microtiter plate were washed with buffered PBS, and 1 mL of fresh DMEM complete nutrient medium (supplemented with 10% FBS, antibiotic-free) containing the plant extracts was added to each well. Bacterial cultures were introduced to achieve a multiplicity of infection (MOI) of a 5:1 bacteria-to-cell ratio, followed by a three-hour incubation at 37 °C. Post-incubation, the medium was removed, and the cells were washed thrice with tempered PBS. For cell lysis and detachment of adhered bacteria, 100 μL of 1% Triton X-100 was added, and after a 10 min incubation at room temperature, 900 μL of LB medium was added. The well contents were homogenized with a micropipette. Dilutions of adherent bacteria and bacterial inoculum were seeded on an LA medium, and after overnight incubation at 37 °C, grown colonies were counted. The negative control contained ethanol instead of the sample.

#### 4.9.2. Invasion Assay

After achieving the MOI and incubating the plates (3 h, 37 °C, as explained in Section 4.9.1), the adhered cells were washed with a medium containing 300 μg/mL gentamicin, followed by additional incubation (1 h, 37 °C). Post-medium removal, the cells were washed with PBS to eliminate bacteria and lysed with 1% Triton X-100. The collected lysate was centrifuged at 4000× *g* rpm (5 min), and the cells were rinsed with 1 × PBS to remove bacteria. Additionally, 10 μL of bacterial dilution (1 × 10^−1^, 1 × 10^−2^, 1 × 10^−3^, and 1 × 10^−4^ in 1 × PBS) was seeded on a TSA medium and incubated for 24 h at 37 °C. Colony counts were conducted, and the number of bacteria per milliliter was calculated. The negative control contained ethanol instead of the sample.

### 4.10. Assessment of Lamiaceae Extracts’ Impact on P. aeruginosa PAO1 Biofilm Formation

Selected Lamiaceae extracts’ impact on *P. aeruginosa* PAO1 biofilm formation was assessed by the crystal-violet method [88]. In the first microtiter plate wells, 200 µL of MIC concentration of extracts (¼MIC for *H. officinalis* and *O. basilicum*, Table 6) was added, after which 100 µL of MHB medium was added to the remaining wells. A concentration gradient was created by transferring 100 µL from row to row, testing MIC and sub-MIC concentrations. An amount of 100 µL of *P. aeruginosa* PAO1 culture, diluted to an OD_600_ of 0.05 (10^7^ CFU/mL), was added to all wells, resulting in double dilutions of the extracts. After a 24 h incubation at 37 °C, the medium was removed, and the biofilm remained. To remove residual planktonic bacteria, the biofilm was washed with sterile distilled water twice, followed by drying (15 min, 25 °C). Biofilm cells were fixed with methanol (10 min) and dried, and each well was stained with 125 µL of 0.1% crystal violet (15 min) for biofilm mass determination. Post-staining, the plates were washed with distilled water twice and left to dry for 10 min. To dissolve residual dye, 200 μL of absolute ethanol was added to each well. After mixing, absorbance was read at 570 nm on a Multiskan FC microtiter plate reader (Thermo Scientific, Waltham, MA, USA). The values were compared with streptomycin (positive control), and the results were presented as the percentage of viable bacteria in the newly formed biofilm.

### 4.11. Assessment of Lamiaceae Extracts Impact on Pre-Existing P. aeruginosa PAO1 Biofilm

The impact of the extract on pre-existing *P. aeruginosa* PAO1 biofilm was assessed following the protocol by Merritt et al. [89]. Overnight culture of *P. aeruginosa* PAO1 was diluted to OD_600_ = 0.01 in an MHB nutrient medium, and 200 µL of suspension was added to each well and incubated for 6 h at 37 °C. After incubation, the microtiter plate was washed twice with sterile PBS buffer to eliminate planktonic bacteria. After drying, 100 µL of MHB nutrient medium containing plant extracts (2 × MIC, except for *H. officinalis* and *O. basilicum*, where ½MIC was used; Table 6) was added, as prepared in the same concentration as described earlier (Section 4.10). Following a 24 h incubation at 37 °C, plate contents were removed, and each well was stained with 125 µL of 0.1% crystal violet for 15 min. The results were evaluated using the same procedure as described earlier (Section 4.10).

### 4.12. Statistical Analyses

For the effects of solvents on total coumarin and triterpene contents, one-way ANOVA and Tukey’s post hoc tests were used, while the *t*-test assessed the significance of results in the adhesion and invasion assays. A statistical analysis was performed with the computer software IBM SPSS v29. The distribution between samples in the assays applied for the determination of biofilm formation and degradation was examined by the Shapiro–Wilk test, and the *t*-test was used to determine statistical significance between the samples. A significance level of *p* < 0.05 was set for all relationships. Pearson’s correlation coefficients (r) represented correlations among total coumarin, triterpene contents, and antibacterial assays, interpreted following Taylor [24]. Integrated Biomarker Response (IBR) values were calculated only for the ethanolic extracts and the results were standardized according to Beliaeff and Burgeot [90]. The results were standardized for the extracts that underwent testing in all experiments (12 ethanol extracts in total). The results of the biofilm formation assay were standardized for the concentration of extracts of 625 μg/mL, while the results of the biofilm degradation were standardized for the concentration of extracts of 312 μg/mL.

## 5. Conclusions

Amidst the consistent increase in bacterial resistance to existing antibiotics and the declining effectiveness of these antibiotics against common infections, our study delved into the antibacterial potential of methanolic, ethanolic, and aqueous extracts derived from 18 aromatic and medicinal Lamiaceae species from Serbia. The effects of these extracts on the growth of Gram-positive and Gram-negative bacteria, *P. aeruginosa* PAO1 invasion of human cells and *P. aeruginosa* PAO1 biofilm formation and degradation were investigated. The results showed that the aqueous extracts had the highest extraction yield. A Pearson’s correlation analysis revealed significant correlations between the phytochemical content of the extracts, especially coumarins, and the inhibition of *B. subtilis* and *P. aeruginosa* growth, suggesting that coumarins may be key contributors to the antibacterial activity of the Lamiaceae extracts. Although none of the extracts inhibited the adhesion *of P. aeruginosa* PAO1, all ethanolic extracts, except those of *H. officinalis* and *M. officinalis*, significantly reduced the number of bacteria capable of invading MRC-5 cells. Furthermore, ethanolic extracts of *O. vulgare*, *H. officinalis*, *M. officinalis*, *M. piperita*, *R. officinalis*, *S. montana*, *T. chamaedrys*, *T. montanum*, *Th. serpyllum*, and *Th. vulgaris* significantly inhibited biofilm formation at certain concentrations. For pre-existing biofilm degradation, *T. chamaedrys* was the most effective, while *H. officinalis*, *M. piperita*, *O. basilicum*, *O. majorana*, *O. vulgare*, *S. montana*, *T. montanum*, *Th. serpyllum*, and *Th. vulgaris* showed significant degradation at lower concentrations. The IBR analysis identified *S. montana* and *O. vulgare* ethanolic extracts as the most promising antibacterial agents, highlighting their potential in defending against bacterial infections. Our findings demonstrate the potent antibacterial efficacy of Lamiaceae extracts and, for the first time, their ability to inhibit *P. aeruginosa* PAO1 invasion of MRC-5 cells, a critical event for bacterial survival and infection initiation. With previous studies primarily focusing on the antibacterial activity of Lamiaceae essential oils, our research underscores the potential of Lamiaceae extracts as powerful natural defenses against multiple aspects of bacterial infections. Further research is needed to identify the specific compounds responsible for the antibacterial activity and to elucidate their mechanisms of action.

## Figures and Tables

**Figure 1 plants-13-01616-f001:**
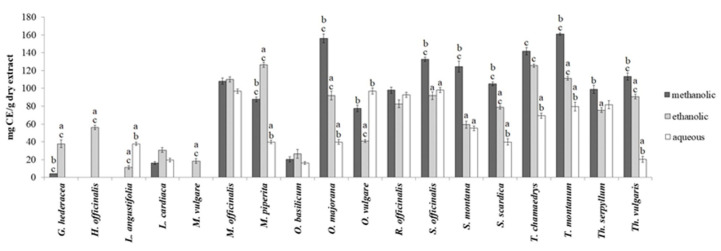
Total coumarin content of methanolic, ethanolic, and aqueous extracts of Lamiaceae representatives. Mean values from different extracts of a plant species, denoted by distinct superscript letters (a–c), show significant differences: a vs. methanolic, b vs. ethanolic, and c vs. aqueous extract (one-way ANOVA, Tukey’s post hoc test, *p* < 0.05).

**Figure 2 plants-13-01616-f002:**
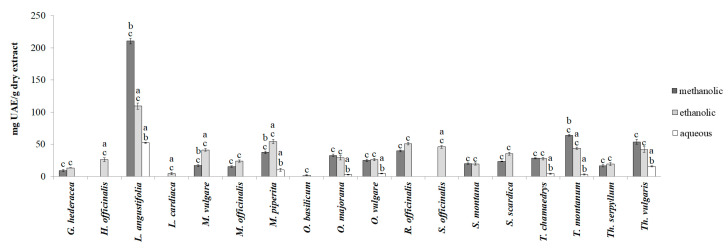
Total triterpene content of methanolic, ethanolic, and aqueous extracts of Lamiaceae representatives. Mean values from different extracts of a plant species, denoted by distinct superscript letters (a–c), show significant differences: a vs. methanolic, b vs. ethanolic, and c vs. aqueous extract (one-way ANOVA, Tukey’s post hoc test, *p* < 0.05).

**Figure 3 plants-13-01616-f003:**
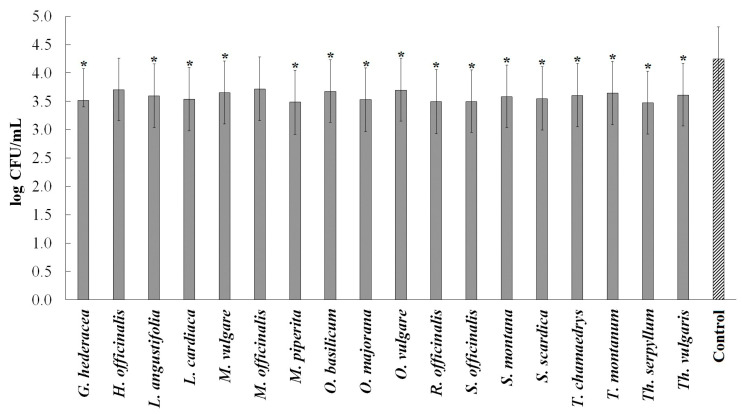
The effect of ethanolic extracts on the invasion of *Pseudomonas aeruginosa* PAO1 in the infection of lung fibroblasts (MRC-5 cells), presented by mean ± standard deviation of log CFU/mL. The values marked with * differ significantly from those of the control (*t*-test, *p* < 0.05).

**Figure 4 plants-13-01616-f004:**
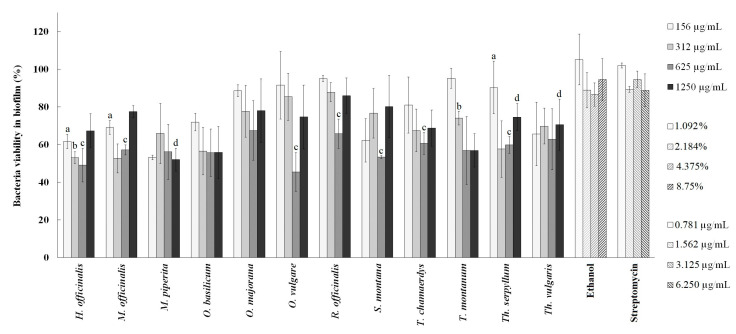
The effect of ethanolic extracts on *Pseudomonas aeruginosa* PAO1 biofilm formation. Statistically significant differences (*p* < 0.05) between the samples and the negative control (ethanol) are represented by the letters a, b, c, and d for each concentration of extract and ethanol.

**Figure 5 plants-13-01616-f005:**
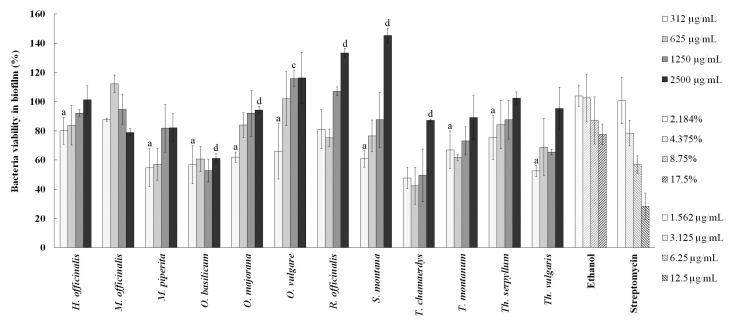
The effect of ethanolic extracts on *Pseudomonas aeruginosa* PAO1 biofilm degradation. Statistically significant differences (*p* < 0.05) between the samples and the negative control (ethanol) are represented by the letters a, b, c, and d for each concentration of extract and ethanol.

**Figure 6 plants-13-01616-f006:**
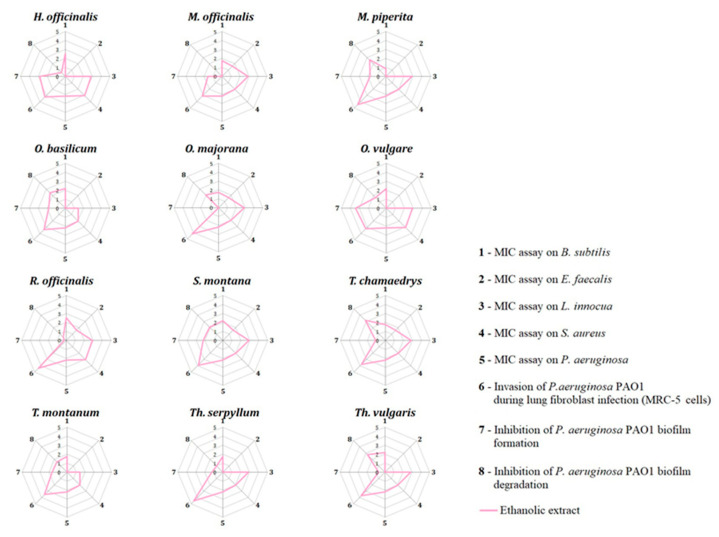
Radial diagrams from IBR analysis: antibacterial activity of ethanolic extracts from selected Lamiaceae representatives.

**Figure 7 plants-13-01616-f007:**
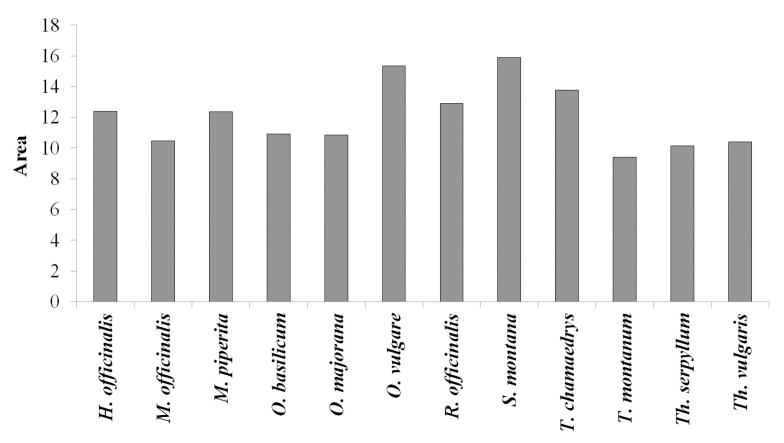
Summarized area of radial diagrams in Figure 6: A larger area indicates a higher antibacterial potential of the tested ethanolic extracts.

**Table 1 plants-13-01616-t001:** Yield of extracts of the examined Lamiaceae representatives.

Plant Species	Yield of Extracts (%)		
	Methanolic	Ethanolic	Aqueous
*G. hederacea*	16.88	13.11	9.07
*H. officinalis*	14.12	10.59	16.63
*L. angustifolia*	20.11	18.05	31.06
*L. cardiaca*	14.27	32.09	18.15
*M. vulgare*	34.10	16.77	14.70
*M. officinalis*	16.10	16.70	23.60
*M. piperita*	30.88	18.20	28.27
*O. basilicum*	28.52	9.92	17.80
*O. majorana*	13.48	14.28	19.68
*O. vulgare*	21.68	13.93	20.73
*R. officinalis*	12.15	10.74	17.98
*S. officinalis*	18.01	21.91	21.84
*S. montana*	11.43	11.31	15.32
*S. scardica*	12.44	10.58	11.25
*T. chamaedrys*	15.43	16.20	16.75
*T. montanum*	15.50	13.64	14.66
*Th. serpyllum*	9.62	9.39	9.72
*Th. vulgaris*	10.82	11.27	12.42

**Table 2 plants-13-01616-t002:** Minimum inhibitory concentrations (MIC) and minimum bactericidal concentrations (MBC) of extracts of Lamiaceae representatives.

Plant Species	Extract	Gram-Positive Bacteria								Gram-Negative Bacteria	
		*B. subtilis*		*E. faecalis*		*L. innocua*		*S. aureus*		*P. aeruginosa*	
		MIC	MBC	MIC	MBC	MIC	MBC	MIC	MBC	MIC	MBC
*G. hederacea*	methanolic	1000	>1000	>1000	nd	>1000	nd	500	1000	>1000	nd
	ethanolic	500	1000	>1000	nd	500	nd	500	1000	500	1000
*H. officinalis*	methanolic	1000	>1000	>1000	nd	>1000	nd	500	1000	>1000	nd
	ethanolic	31.25	62.50	>1000	nd	500	nd	250	500	250	500
*L. angustifolia*	methanolic	250	500	1000	nd	1000	nd	500	1000	>1000	nd
	ethanolic	250	500	>1000	nd	500	nd	500	1000	500	1000
*L. cardiaca*	methanolic	250	500	1000	>1000	1000	nd	500	1000	>1000	nd
	ethanolic	125	250	1000	nd	500	nd	250	500	500	1000
*M. vulgare*	methanolic	>1000	nd	>1000	nd	>1000	nd	1000	>1000	>1000	nd
	ethanolic	250	500	>1000	nd	500	1000	250	500	500	1000
*M. officinalis*	methanolic	500	1000	1000	>1000	1000	nd	250	500	>1000	nd
	ethanolic	500	1000	1000	nd	500	nd	500	1000	250	500
*M. piperita*	methanolic	500	1000	>1000	nd	1000	nd	500	1000	>1000	nd
	ethanolic	1000	nd	>1000	nd	500	1000	500	1000	250	500
*O. basilicum*	methanolic	500	1000	1000	>1000	1000	nd	500	1000	>1000	nd
	ethanolic	250	500	>1000	nd	1000	nd	500	1000	250	500
*O. majorana*	methanolic	250	500	500	1000	500	1000	500	1000	>1000	nd
	ethanolic	500	1000	1000	>1000	500	1000	500	1000	250	500
*O. vulgare*	methanolic	500	1000	1000	nd	500	1000	500	1000	>1000	nd
	ethanolic	250	500	>1000	nd	500	1000	250	500	250	500
*R. officinalis*	methanolic	125	250	500	1000	250	500	500	1000	>1000	nd
	ethanolic	62.50	125	1000	nd	500	1000	250	500	250	500
*S. officinalis*	methanolic	500	1000	1000	nd	1000	nd	500	1000	>1000	nd
	ethanolic	62.50	125	500	nd	250	500	125	250	500	1000
*S. montana*	methanolic	500	1000	1000	nd	500	1000	1000	>1000	>1000	nd
	ethanolic	250	500	1000	nd	500	nd	500	1000	250	500
*S. scardica*	methanolic	31.25	62.50	1000	>1000	500	1000	250	500	>1000	nd
	ethanolic	125	250	1000	>1000	500	1000	250	500	250	500
*T. chamaerdys*	methanolic	500	1000	1000	nd	500	1000	1000	>1000	>1000	nd
	ethanolic	500	1000	1000	>1000	500	nd	500	1000	250	500
*T. montanum*	methanolic	500	1000	1000	>1000	1000	nd	1000	>1000	>1000	nd
	ethanolic	500	1000	>1000	nd	1000	nd	500	1000	250	500
*Th. serpyllum*	methanolic	500	1000	1000	>1000	500	1000	1000	>1000	>1000	nd
	ethanolic	500	1000	>1000	nd	500	nd	500	1000	250	500
*Th. vulgaris*	methanolic	250	500	1000	>1000	250	500	1000	>1000	>1000	nd
	ethanolic	250	500	1000	>1000	500	1000	500	1000	250	500
Positive controls											
Streptomycin		≤3.125	3.125	≤200	200	≤25	25	≤1.532	3.125	25	50
Rifampicin		-	-	≤3.125	50	≤3.906	7.812	-	-	-	-

nd—not detected; the results are presented as µg/mL.

**Table 3 plants-13-01616-t003:** The inter-correlation of assays for total coumarin and triterpene contents with antibacterial activity.

	Total Coumarin Content	Total Triterpene Content	MIC for *B. subtilis*	MIC for *E. faecalis*	MIC for *L. innocua*	MIC for *S. aureus*	MIC for *P. aeruginosa* ATCC 15442	Invasion of *P. aeruginosa* PAO1	Inhibition of Biofilm Formation	Inhibition of Biofilm Degradation
Total coumarin content	1	−0.01	**−0.50**	0.34	0.06	−0.24	**0.56**	0.20	0.10	0.43
Total triterpene content		1	−0.03	−0.11	0.23	0.00	−0.26	0.15	−0.27	−0.20
MIC for *B. subtilis*			1	0.30	0.16	**0.68**	−0.24	−0.19	−0.13	−0.44
MIC for *E. faecalis*				1	**0.47**	0.39	−0.10	0.33	−0.36	−0.20
MIC for *L. innocua*					1	0.42	−0.33	0.39	−0.29	−0.34
MIC for *S. aureus*						1	−0.33	0.04	−0.12	**−0.53**
MIC for *P. aeruginosa*							1	−0.22	**0.60**	**0.63**
Invasion of *P. aeruginosa* PAO1								1	**−0.57**	−0.09
Inhibition of biofilm formation									1	0.44
Inhibition of biofilm degradation										1

r—Pearson’s correlation coefficients; |r| < 0.35 weak correlation; 0.35 < |r| < 0.67 moderate correlation; 0.68 < |r| < 1 strong correlation [24]. The correlations highlighted in bold are significant at *p* < 0.05.

**Table 4 plants-13-01616-t004:** Plant species used in this study and their respective voucher numbers.

Plant Species	Voucher Number
*Glechoma hederacea* L.	302181
*Hyssopus officinalis* L.	302211
*Lavandula angustifolia* Mill.	300071
*Leonurus cardiaca* L.	302221
*Marrubium vulgare* L.	302241
*Melissa officinalis* L.	301121
*Mentha*×*piperita* L.	301131
*Ocimum basilicum* L.	302061
*Origanum majorana* L.	302231
*Origanum vulgare* L.	302291
*Rosmarinus officinalis* L.	301211
*Salvia officinalis* L.	301241
*Satureja montana* L. *s.l.*	302311
*Sideritis scardica* L.	302331
*Teucrium chamaedrys* L.	302091
*Teucrium montanum* L.	302351
*Thymus serpyllum* L.	302321
*Thymus vulgaris* L.	302361

**Table 5 plants-13-01616-t005:** Bacterial strains used in the experimental work.

Gram-Positive Strains	Gram-Negative Strains
*Bacillus subtilis* ATCC 6633	*Escherichia coli* ATCC 25922
*Enterococcus faecalis* ATCC 29212	*Pseudomonas aeruginosa* ATCC 15442
*Listeria innocua* ATCC 33090	*Pseudomonas aeruginosa* PAO1 ATCC 15692
*Staphylococcus aureus* ATCC 25923	*Salmonella enterica* subsp. *enterica* serovar Typhimurium ATCC 14028

**Table 6 plants-13-01616-t006:** Minimum inhibitory concentrations (MIC) and minimum bactericidal concentrations (MBC) of ethanolic extracts and positive control against *Pseudomonas aeruginosa* PAO1.

Plant Species	MIC	MBC
*H. officinalis*	5000	>5000
*M. officinalis*	1250	2500
*M. piperita*	1250	2500
*O. basilicum*	5000	>5000
*O. majorana*	1250	2500
*O. vulgare*	1250	2500
*R. officinalis*	1250	2500
*S. montana*	1250	2500
*T. chamaedrys*	1250	2500
*T. montanum*	1250	2500
*Th. serpyllum*	1250	2500
*Th. vulgaris*	1250	2500
Positive control		
Streptomycin	6.25	12.5

The results are presented as μg/mL.

## Data Availability

The original contributions presented in this study are included in the article; further inquiries can be directed to the corresponding author.

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
