# Peer review of "Exploring the Antibacterial Potential of Lamiaceae Plant Extracts: Inhibition of Bacterial Growth, Adhesion, Invasion, and Biofilm Formation and Degradation in Pseudomonas aeruginosa PAO1"

_plants, 2024, doi:10.3390/plants13121616_

Round 1

Reviewer 1 Report

Comments and Suggestions for Authors

The resistance of pathogenic microorganisms to antibiotics is a global concern, and plants serve as an important source of new antimicrobial agents. This manuscript systematically investigates the antibacterial potential of extracts from 18 aromatic and medicinal Lamiaceae plant species native to Serbia. The study examines the effects of methanol and ethanol extracts on bacterial growth, Pseudomonas aeruginosa PAO1 adhesion and invasion, and biofilm formation and degradation. The manuscript substantiates the ethanolic extracts of S. montana and O. vulgare as the most promising antibacterial agents, with potential roles in defending against bacterial infections. The manuscript is well-written, the research is comprehensive, and the conclusions are correct. It can be accepted for publication after minor revisions. There are a few improvements needed in the manuscript:

1. Standard deviations need to be added to the data in Table 1.

2. The significance test results and annotations in Figures 1 and 2 are confusing and require restatement and clearer markings.

3. Significance test results need to be added to Figures 4, 5, and Table 3.

4. There is no significant correlation between total coumarin and triterpene content with MIC and biofilm, the authors need to discuss these results further.

5. The extraction yield for both methanol and ethanol are low. The current extraction methods in the manuscript lack references, and discussing potential improvements for extraction yield in the future is warranted.

6. The current manuscript contains an excessive number of references which need to be streamlined.

Author Response

Dear Reviewer,

I am writing to you regarding the manuscript entitled ”Natural shield: Unveiling the antibacterial efficacy of Lamiaceae plant extracts in the inhibition of bacterial growth, Pseudomonas aeruginosa PAO1 adhesion and invasion, and biofilm formation and degradation“, Manuscript ID: plants-3059728. The manuscript was carefully checked and corrected according to your remarks.

We would like to thank you for all the suggestions and comments, which significantly contributed to the improvement of our manuscript. We have made changes in the manuscript (highlighted in green) as follows:

  1. Standard deviations need to be added to the data in Table 1.
    • Response: I regret to inform you that we cannot add standard deviations to the extraction yield data, as the plant extracts were obtained from a single extraction process. Consequently, there is no replication to calculate the standard deviations. In many cases in the literature, extraction yield percentages are reported as single values, especially when the focus is on the comparative efficiency of different solvents rather than the variability within a single method.
  2. The significance test results and annotations in Figures 1 and 2 are confusing and require restatement and clearer markings.
    • Response: We have added a more detailed explanation of the statistics in Figures 1 and 2 (lines 126-129, and 141-143). Specifically, we clarified that mean values obtained from different extracts of the same plant species are marked with distinct superscript letters (a–c), indicating significant differences: ’a’ compared to methanolic extract, ’b’ compared to ethanolic extract, and ’c’ compared to aqueous extract (one-way ANOVA, Tukey post-hoc test, p<0.05). We hope that the annotations and explanations under Figures 1 and 2 are now clearer.
  3. Significance test results need to be added to Figures 4, 5, and Table 3.
    • Response: We appreciate your suggestions which have greatly improved the quality of our manuscript. We have added the statistical significance test results to Figures 4, 5, and Table 3. Each figure and table now includes annotations indicating significant differences, following the same methodology used in previous figures. We have expanded our manuscript to address these results in detail. This should provide a clearer understanding of the statistical analysis and the significant differences observed in our data.

However, due to the large number of samples exhibiting relatively similar effects, a comparative analysis to determine the significance of differences between individual extracts in Figures 4 and 5 was not conducted. This decision was also made to avoid further complicating the discussion. Additionally, the positive control, the antibiotic streptomycin, was tested at significantly lower concentrations, making statistical comparison between the results for the extracts and those for streptomycin impractical. Therefore, statistical analysis was performed to assess the significance of differences between the extracts and the negative control (ethanol) to determine the extent to which the extracts themselves contribute to biofilm inhibition or degradation. The revisions have been incorporated into Figures 4 and 5, as well as into the Results, Discussion, Material and Methods, and Conclusion sections (lines 179-191, 194-196, 203-207, 213-215, 438-445, 631-634, 657-663).

  1. There is no significant correlation between total coumarin and triterpene content with MIC and biofilm, the authors need to discuss these results further.
    • Response: Upon further analysis and the addition of statistical significance data, we have found a significant correlation between the total coumarin content and the MIC assay for certain bacteria. These findings and changes to the existing text have been highlighted in green in the Abstract, Results, Discussion, and Conclusion sections. We have expanded our discussion to address these correlations in detail, providing a deeper insight into the relationship between coumarin content and antibacterial activity. The revisions regarding the correlation are highlighted in lines 25, 224-251, 366-368, 652-655.
  2. The extraction yield for both methanol and ethanol are low. The current extraction methods in the manuscript lack references, and discussing potential improvements for extraction yield in the future is warranted.
    • Response: We have added a paragraph to the Discussion section (lines 295-310), supported by two additional references. This paragraph emphasizes the importance of integrating advanced techniques and optimizing parameters to improve extraction yields in future research.
  3. The current manuscript contains an excessive number of references which need to be streamlined.
    • Response: We have streamlined our reference list by omitting nine references. However, during the revision process, we needed to add four new references to ensure the accuracy and completeness of our study (lines 698-699, 704-705, 767-771). The primary reason for the extensive number of references is that our study investigates 18 plant species and various extracts. Previous research has predominantly focused on individual Lamiaceae plants or smaller groups of plants, necessitating a broader range of references to support the comprehensive scope of our research. This variety and volume of research objects inherently require more references to provide a thorough and well-supported analysis.

Reviewer 2 Report

Comments and Suggestions for Authors

Recommendation for revision:

Query#1

I suggest to the authors to revise the title as follows: “Exploring the Antibacterial Potential of Lamiaceae Plant Extracts: Inhibition of Bacterial Growth, Adhesion, Invasion, and Biofilm Formation and Degradation in Pseudomonas aeruginosa PAO1"

Query#2

The introduction was noted to be overly concise and lacking in sufficient detail. Although the authors discussed the significance of biofilm bacterial infections, I suggest that the authors highlight that different multi-resistant strains belonging to species Enterococcus faecium, Staphylococcus aureus, Klebsiella pneumoniae, Acinetobacter baumannii, Pseudomonas aeruginosa, and Enterobacter spp., acronymically known as the “ESKAPE” pathogens, are representative examples of pathogens responsible for serious and difficult-to-treat nosocomial infections due to their antibiotic resistance.

This additional context will provide a more comprehensive foundation for understanding the importance of biofilm formation.

To support this suggestion, I recommend that the authors cite the updated articles provided:

Int J Mol Sci. 2023;24(5):4872. doi:10.3390/ijms24054872

Front Cell Infect Microbiol. 2023;13:1159798. doi:10.3389/fcimb.2023.1159798

Query#3

In the table 6 please revise "streptomyocin" in "streptomycin"

Comments on the Quality of English Language

Double check the entire documents, different typos are present.

Review the English form of the whole article different parts need careful checking of the English form and language.

Author Response

Dear Reviewer,

I am writing to you regarding our manuscript entitled “Natural shield: Unveiling the antibacterial efficacy of Lamiaceae plant extracts in the inhibition of bacterial growth, Pseudomonas aeruginosa PAO1 adhesion and invasion, and biofilm formation and degradation”, Manuscript ID: plants-3059728.

We have carefully reviewed and revised the manuscript in accordance with your remarks. We would like to extend our gratitude for your valuable suggestions and comments, which have significantly enhanced the quality of our manuscript. The changes made to the manuscript are highlighted in green and include the following:

  1. English very difficult to understand/incomprehensible.
  2. Double check the entire documents, different typos are present. Review the English form of the whole article different parts need careful checking of the English form and language.

Response: Thank you for your feedback. We have thoroughly reviewed and spell-checked the entire manuscript. Additionally, we have revised various sections to improve readability and ensure the language is clear and concise throughout. We believe these changes have addressed the typos and enhanced the overall quality of the manuscript.

  1. I suggest to the authors to revise the title as follows: “Exploring the Antibacterial Potential of Lamiaceae Plant Extracts: Inhibition of Bacterial Growth, Adhesion, Invasion, and Biofilm Formation and Degradation in Pseudomonas aeruginosa PAO1".

Response: Thank you for your suggestion regarding the title. We have revised the title as follows: “Exploring the Antibacterial Potential of Lamiaceae Plant Extracts: Inhibition of Bacterial Growth, Adhesion, Invasion, and Biofilm Formation and Degradation in Pseudomonas aeruginosa PAO1“.

  1. The introduction was noted to be overly concise and lacking in sufficient detail. Although the authors discussed the significance of biofilm bacterial infections, I suggest that the authors highlight that different multi-resistant strains belonging to species Enterococcus faecium, Staphylococcus aureus, Klebsiella pneumoniae, Acinetobacter baumannii, Pseudomonas aeruginosa, and Enterobacter, acronymically known as the “ESKAPE” pathogens, are representative examples of pathogens responsible for serious and difficult-to-treat nosocomial infections due to their antibiotic resistance.

This additional context will provide a more comprehensive foundation for understanding the importance of biofilm formation.

To support this suggestion, I recommend that the authors cite the updated articles provided:

Int J Mol Sci. 2023;24(5):4872. doi:10.3390/ijms24054872

Front Cell Infect Microbiol. 2023;13:1159798. doi:10.3389/fcimb.2023.1159798.

Response: Thank you for your valuable feedback regarding the introduction of our manuscript. We have revised the introduction to provide more detailed information about the significance of biofilm bacterial infections, particularly highlighting the role of multi-resistant strains such as Enterococcus faecium, Staphylococcus aureus, Klebsiella pneumoniae, Acinetobacter baumannii, Pseudomonas aeruginosa, and Enterobacter spp., collectively known as the “ESKAPE” pathogens. We have incorporated this additional context to offer a more comprehensive understanding of the importance of biofilm formation (the revisions have been highlighted in green within lines 38-62, and 88-89).

  1. In the table 6 please revise "streptomyocin" in "streptomycin".

Response: Thank you for this remark. We have corrected this typo.

Reviewer 3 Report

Comments and Suggestions for Authors

The presented study brings several interesting results and is, in my opinion, well-conceived overall. From the study, it is not entirely clear why only one representative of G-negative bacteria (see T2) and one strain was chosen for testing?

However, I have several comments and suggestions for changes to the text; some parts of the text or object are not completely clear or properly presented.

1. L454 - temperature without "+"

2. Table 5, L148, etc. - the correct taxonomic designation of Salmonella must be given

3. L499 - gap under table 5

4. L508 - it is better to write information about cell concentration in a sentence (simultaneously without using "=").

5. L521, L562, .... etc. - (CFU/mL = 105) → 105 CFU/mL!

6. Table 6 - place the name "positive control" appropriately within the table

7. L153 - The sentence "The results....) would be more appropriately placed in the title of the object (essential information), not under the object, and I do not recommend this sentence combination at all after the previous explanation of the table.

8. Figure 3, etc. - correctly write the axis name as "log CFU/mL"!

9. Is the Fig 7 object indispensable? Would it not be possible, for example, to add data to object Fig 6 (only numerical data)?

Author Response

Dear Reviewer,

I am reaching out regarding our manuscript “Natural shield: Unveiling the antibacterial efficacy of Lamiaceae plant extracts in the inhibition of bacterial growth, Pseudomonas aeruginosa PAO1 adhesion and invasion, and biofilm formation and degradation” (Manuscript ID: plants-3059728).

We have meticulously reviewed and revised the manuscript based on your insightful remarks. We sincerely appreciate your valuable suggestions and comments, which have greatly improved the quality of our work. The revisions made are highlighted in green, and our responses to your queries are listed below:

  1. The presented study brings several interesting results and is, in my opinion, well-conceived overall. From the study, it is not entirely clear why only one representative of G-negative bacteria (see T2) and one strain was chosen for testing?

Response: In the study, we assessed bacterial growth (minimum inhibitory concentration - MIC) on four Gram-positive bacteria (Bacillus subtilis, Enterococcus faecalis, Listeria innocua, and Staphylococcus aureus) and three Gram-negative bacteria (Escherichia coli, Pseudomonas aeruginosa, and Salmonella typhimurium). However, we did not include the data for E. coli and S. typhimurium in Table 2 because the minimum inhibitory concentration (MIC) for all tested extracts against these two bacteria was higher than 1000 µg/mL, indicating a lack of significant inhibitory effect, which was similar to the exclusion of data for aqueous extracts which also showed MIC values higher than 1000 µg/mL.

We have now added the following text to the Results section for better clarity (lines 147-149):

„Additionally, results for the inhibition of E. coli and S. typhimurium growth are not provided in Table 2, as their MICs and MBCs exceeded the highest tested concentration of extracts, 1000 µg/mL“.

  1. I have several comments and suggestions for changes to the text; some parts of the text or object are not completely clear or properly presented.

Response: Thank you for your comments and suggestions regarding the clarity and presentation of certain parts of our manuscript. We have thoroughly reviewed the text and performed a comprehensive spell check to ensure accuracy and clarity.

  1. L454 - temperature without "+"

Response: We have corrected this and removed the "+" sign to ensure proper formatting.

  1. Table 5, L148, etc. - the correct taxonomic designation of Salmonella must be given

Response: We have updated Table 5 to include the correct taxonomic designation: „Salmonella enterica subsp. enterica serovar Typhimurium ATCC 14028“ (line 539). Additionally, we have ensured consistency by adding this designation in the introduction section where the name first appears (line 84).

  1. L499 - gap under table 5

Response: We have corrected the formatting.

  1. L508 - it is better to write information about cell concentration in a sentence (simultaneously without using "=").

Response: We have revised the text to include this information in a sentence format, removing the use of "=".

  1. L521, L562, .... etc. - (CFU/mL = 105) → 105 CFU/mL!

Response: We have corrected these to the proper format (105 CFU/mL) throughout the manuscript.

  1. Table 6 - place the name "positive control" appropriately within the table

Response: We have appropriately placed the name "positive control" within the table as advised.

  1. L153 - The sentence "The results....) would be more appropriately placed in the title of the object (essential information), not under the object, and I do not recommend this sentence combination at all after the previous explanation of the table.

Response: We have removed the sentences "The results are shown as mg CE/g, for a concentration of 500 μg/mL" and "The results are shown as mg UAE/g, for a concentration of 500 μg/mL" from under Figures 1 and 2.

  1. Figure 3, etc. - correctly write the axis name as "log CFU/mL"!

Response: We have corrected the axis name to "log CFU/mL" as recommended (Figure 3 and line 170).

  1. Is the Fig 7 object indispensable? Would it not be possible, for example, to add data to object Fig 6 (only numerical data)?

Response: Figure 7 is necessary as it represents the final result of the IBR analysis, which cannot be adequately conveyed by adding numerical data to Figure 6 alone. The visual representation in Figure 7 is crucial for illustrating the overall outcomes and conclusions of our study.